# Balloon Pulmonary Angioplasty in Patients with Chronic Thromboembolic Pulmonary Hypertension: Impact on Clinical and Hemodynamic Parameters, Quality of Life and Risk Profile

**DOI:** 10.3390/jcm9113608

**Published:** 2020-11-09

**Authors:** Pavel Jansa, Samuel Heller, Michal Svoboda, Michal Pad’our, David Ambrož, Vladimír Dytrych, Michal Širanec, Tomáš Kovárník, Marián Felšőci, Martin Hutyra, Aleš Linhart, Jaroslav Lindner, Michael Aschermann

**Affiliations:** 12nd Department of Medicine–Department of Cardiovascular Medicine, First Faculty of Medicine, Charles University and General University Hospital, 128 08 Prague, Czech Republic; shel@lf1.cuni.cz (S.H.); michal.padour@vfn.cz (M.P.); david.ambroz@vfn.cz (D.A.); vladimir.dytrych@vfn.cz (V.D.); michal.siranec@vfn.cz (M.Š.); tomas.kovarnik@vfn.cz (T.K.); alinh@lf1.cuni.cz (A.L.); michael.aschermann@vfn.cz (M.A.); 2Institute of Biostatistics and Analyses, Faculty of Medicine, Masaryk University, 625 00 Brno, Czech Republic; svoboda@biostatistika.cz; 3Department of Internal Medicine and Cardiology, University Hospital Brno and Faculty of Medicine, Masaryk University, 625 00 Brno, Czech Republic; Felsoci.Marian@fnbrno.cz; 4Department of Internal Medicine I–Cardiology, University Hospital, 779 00 Olomouc, Czech Republic; martin.hutyra@fnol.cz; 52nd Department of Surgery–Department of Cardiovascular Surgery, First Faculty of Medicine, Charles University and General University Hospital, 128 08 Prague, Czech Republic; jaroslav.lindner@vfn.cz

**Keywords:** chronic thromboembolic pulmonary hypertension, balloon pulmonary angioplasty, efficacy, safety, quality of life, risk profile, survival

## Abstract

Balloon pulmonary angioplasty (BPA) is a novel treatment option for patients with chronic thromboembolic pulmonary hypertension (CTEPH) who are not eligible for pulmonary endarterectomy (PEA) or suffer from persistent pulmonary hypertension after PEA. The aim of this study was to evaluate the real-life efficacy and safety of BPA in a consecutive group of patients who were diagnosed and treated in the national referral center for CTEPH in the Czech Republic. Here we report data from 160 BPA procedures performed in 64 patients. Efficacy analysis was performed in the subgroup of 25 patients who completed BPA series. Significant improvements were observed in New York Heart Association functional class (4% to 79% in I/II, *p* < 0.001), 6 min walking test distance (+54.3 m, *p* < 0.001), risk profile (15.8% to 68.5% with presence of 2/3 low risk criteria, *p* < 0.001), pulmonary artery mean pressure (−18%, *p* < 0.001), pulmonary vascular resistance (−32%, *p* < 0.001), stroke volume (+17%, *p* = 0.011) and quality of life (+37% in assessment of overall health status by a patient, *p* < 0.001). We observed 1 fatal periprocedural complication (1.6% of all 64 patients) and 19 BPA-related non-fatal complications (11.9% of all 160 interventions) that predominantly included hemoptysis (10.0% of all sessions). Overall survival at 12 months was 94.6%.

## 1. Introduction

Chronic thromboembolic pulmonary hypertension (CTEPH) is characterized by precapillary pulmonary hypertension (PH) and increased right ventricular strain, leading to right heart failure and premature death [1]. It is a chronic complication of acute pulmonary embolism caused by the formation of persistent thrombotic obstructions in pulmonary vessels and concurrent peripheral vascular remodeling with abnormal angiogenesis, disordered fibrinolysis and endothelial dysfunction. Chronic thromboembolic pulmonary hypertension develops in approximately 2–4% of patients who survive acute pulmonary embolism [2].

Surgical pulmonary endarterectomy (PEA) is an established standard treatment that is indicated for 50–70% of patients with confirmed CTEPH. However, a certain proportion of technically operable patients eventually do not have the surgery since they are either at very high risk or they refuse it. Patients who do not undergo the surgery have significantly worse prognosis in comparison to patients after an executed PEA. Similarly, in approximately 25% of patients after PEA residual PH remains, which is associated with persistent CTEPH symptoms and a deteriorated prognosis [3].

In some inoperable patients and in some patients with residual PH after PEA, pharmacological therapy is successful if peripheral vascular remodeling is the predominant cause. To date, the only approved and widely used drug is riociguat—a soluble guanylate cyclase stimulator [4]. Nevertheless, there are also other specific treatments for pulmonary arterial hypertension (PAH) used off-label in clinical practice.

Balloon pulmonary angioplasty (BPA) is a novel treatment option for CTEPH patients not eligible for PEA with the presence of vascular lesions such as ring-like stenosis lesions, web lesions, subtotal lesions, total occlusion lesions or tortuous lesions in consequence of blood clot formation [5]. Balloon pulmonary angioplasty is an endovascular procedure that involves dilatation performed at various segments during repeated sessions. The BPA procedure should be performed exclusively at centers specialized in complex CTEPH management, including experience with treatment of periprocedural and postprocedural complications. Lung injuries with or without hypoxemia and/or hemoptysis are the main complications of BPA [6].

The results of BPA treatment from a cohort of 18 patients were published in 2001 for the first time [7]. The authors achieved a mild amelioration of hemodynamics parameters at the cost of a relatively high occurrence of potentially fatal complications. Recently, several Japanese centers published results of BPA in smaller patient populations with greater improvements in efficacy and safety, especially because of technique refinement (a smaller balloon relative to the actual vessel size was selected to reduce the risk of complications and restore minimal blood flow to the occluded pulmonary vessels during initial sessions, while a larger balloon size was used to optimize the dilatation during subsequent sessions) [7,8,9]. Ultimately, a decrease of more than 50% in pulmonary vascular resistance (PVR) after BPA series was demonstrated in a multicenter registry from seven Japanese centers consisting of 308 patients [10].

In Europe, the results of BPA treatment in 20 patients were published by Norwegian authors in 2013 [11]. In comparison to Japanese centers, hemodynamic effects were lower while the occurrence of complications, including those that were fatal, were much higher. In recent years, a number of European centers (especially in Poland, Germany, France and the United Kingdom) have initiated BPA programs using the BPA technique refined in Japan [12,13,14,15], and first experiences were published from the Mayo Clinic in Rochester, Minnesota [16].

In the Czech Republic, complex CTEPH management has been centralized at the Center of Pulmonary Hypertension in the General University Hospital in Prague [17]. Between 2003 and 2019, 623 patients with CTEPH had been diagnosed and 360 PEA procedures performed. Riociguat therapy has been available from 2015. We performed the first BPA in 2013 and a few others in the following years. Since 2017, we have systematically employed BPA as a CTEPH treatment option.

The aim of this study was to evaluate the real-life efficacy and safety of BPA in a consecutive group of patients who were diagnosed and treated in the national referral center for CTEPH in the Czech Republic. Here, we present our initial experience with BPA in 64 patients treated at our institution.

## 2. Experimental Section

### 2.1. Study Design

The aim of this study was to evaluate the efficacy and safety of BPA in the initial cohort of patients at our institution, the Center of Pulmonary Hypertension in the General University Hospital in Prague, the Czech Republic. Data were obtained retrospectively from consecutive patients with CTEPH who were indicated for and underwent BPA in real-life settings. The primary objective of the BPA series in this initial stage of the BPA program was to achieve a decrease in mean pulmonary artery pressure (PAMP) of 20% from baseline and in PVR of 30% from baseline.

As source documents we used all applicable medical records in the patient documentation. Data collection was conducted in accordance with the principles laid down in the 18th World Medical Assembly (Helsinki, 1964), including all subsequent amendments, and in compliance with all laws and regulations of the Czech Republic. All patients signed an informed consent form prior enrolment to the study. The approval of retrospective data collection was provided by the ethics committee of the General University Hospital in Prague (ID 1208/18 S-IV). The first BPA was performed on 21 October 2013 and the last on 24 October 2019. Here we report data collected up to 31 October 2019.

### 2.2. Patient Population

Chronic thromboembolic pulmonary hypertension diagnosis was based on clinical, laboratory and imaging assessments in conformity with standard guidelines. In addition, the modality of CTEPH management (PEA, pharmacotherapy, BPA or combination of methods) for individual patients was selected by an interdisciplinary team consisting of a PEA surgeon, PH specialist, cardiac anesthesiologist and an interventional cardiologist experienced in the BPA procedure according to current applicable guidelines. Balloon pulmonary angioplasty was offered to patients with inoperable CTEPH due to peripheral involvement and to patients with persistent PH after PEA. Exceptionally, operable patients who had refused PEA also underwent BPA. Patients with PVR >8 Wood units (WU) initially received pharmacotherapy to suppress peripheral vascular remodeling, to improve hemodynamic endpoints and to lower the risk of periprocedural BPA complications.

Determination of target lesions for BPA procedure was based on angiographic classification established by Kawakami et al. [6] who characterized the following lesion types: Ring-like stenosis, web lesion, subtotal lesion, total occlusion lesion and tortuous lesion.

### 2.3. Assessed Endpoints

Using evaluations of New York Heart Association (NYHA) functional class, the 6 min walk distance test (6MWD) and level of natriuretic peptides, patient risk profiles were determined as proposed by Boucly et al. [18] by the number of present noninvasive low-risk criteria: (1) NYHA functional class I or II, (2) 6MWD >440 m and (3) N-terminal pro-brain natriuretic peptide (NTproBNP) <300 ng/L.

Hemodynamics parameters were evaluated during the right heart catheterization as follows: right atrial mean pressure (RAMP), pulmonary artery pressure (systolic, diastolic and mean) and pulmonary capillary wedge pressure (PCWP) obtained after catheter balloon inflation at the end of expiration; cardiac output (CO) was measured with thermodilution with cold saline; cardiac index (CI), PVR and total pulmonary vascular resistance (TPR) were calculated based on previous measurements. Furthermore, the following parameters describing right ventricular function (RV) were calculated: Stroke volume (SV) (cardiac output/heart rate), stroke volume index (SVI) (cardiac index/heart rate), cardiac efficiency (SV/pulmonary artery mean pressure), pulmonary artery (PA) elastance (pulmonary artery systolic pressure/SV) and compliance (SV/(pulmonary artery systolic pressure-pulmonary artery diastolic pressure)).

To assess patient-reported quality of life we used the EQ-5D-5L questionnaire, which evaluates five aspects of health (i.e., mobility, self-care, usual activities, pain/discomfort and anxiety/depression), with each dimension containing five response levels of severity. The higher the EQ-5D-5L score, the better the state of health [19].

For all endpoints of efficacy as described above, data obtained prior to the first BPA (considered as baseline data) and approximately 3 months after the last BPA were collected. Safety data, including any complication related to BPA procedure and survival, with all causes of death noted, were collected continually up to the data cutoff on 31 October 2019. Furthermore, data from the time of CTEPH diagnosis was also available.

### 2.4. BPA Procedure

Warfarin anticoagulation was discontinued 6 days before the BPA session. A therapeutic dose of low molecular weight heparin (LMWH) was used as a bridging therapy up to one day before BPA. In patients receiving non-vitamin K antagonist oral anticoagulants (NOACs), administration was interrupted one day before the BPA without the LMWH bridge. During the actual procedure, oxygen inhalation was given with continuous monitoring of oxygen saturation, electrocardiogram (ECG), blood pressure and heart rate.

We used predominantly femoral access, but a jugular site was employed for a few interventions as well. At the beginning of each procedure, right heart catheterization was performed using a 7 French sheath and Swan–Ganz thermodilution catheter (B Braun Melsungen AG, Germany) to measure hemodynamic parameters. Then, BPA was initiated by the replacement of the 7 French sheath with a guiding sheath (6 French Destination 65 cm or 6 French Balt 80 cm). A 6 French guiding catheter (Multipurpose, Judkins right 4 or 5, Amplatz left; Boston Scientific, Marlborough, MA, USA) was introduced and unfractionated heparin at a dose of 2500 IU was administered into the guiding catheter. The guide wire (PT2 Moderate Support, Boston Scientific, Marlborough, MA, USA or Asahi Sion Blue, Asahi Intecc Co, Japan) was placed into the target branch. We selected undersized balloons (1.5–4/20 mm, Abbott, Abbott Park, IL, USA) to avoid pulmonary vascular injury. The larger balloons were used in later sessions for postdilatation. The lesions were dilated by multiple balloon inflations at 2–8 segmental or subsegmental branches during one session. We had never exceeded the limit of 300 mL contrast medium and 40 min of total time of skiascopy in any session. A subsequent procedure was planned within 1–2 months. Figure 1 demonstrates selective pulmonary angiograms and BPA result in an inoperable patient with CTEPH.

### 2.5. Statistical Analysis

Analysis of safety and survival was performed in all patients who underwent at least 1 session of BPA. The analysis of efficacy endpoints was conducted only in the subgroup of patient who completed all intended BPA sessions.

Continuous variables were described by valid number (*n*), mean (standard deviation (SD)) and median (5th and 95th percentile). Absolute and relative frequencies were used for a description of categorical variables. Changes in the values of continuous variables over time were evaluated using the Wilcoxon paired test. Overall survival of patients was analyzed by the Kaplan–Meier estimate of survival function. The Kaplan–Meier curve was supplemented by the number of patients at risk and the proportion surviving at analyzed times. The analysis was performed using the software IBM SPSS Statistics, with the level of significance defined as 0.05.

## 3. Results

### 3.1. Patient Characteristics

Overall, 64 patients (50% of females, mean age of 61.4 ± 12.1 years) underwent at least one session of BPA up to 31 October 2019 (Figure 2). In total, we analyzed data from 160 BPA sessions performed at the General University Hospital in Prague. At the time of diagnosis, 25.8% of patients had NYHA class II, 71% had NYHA class III and 3.2% had NYHA class IV. All patients had been anticoagulated for at least 3 months—68.3% by warfarin and 31.7% by NOACs. Median time from diagnosis to the first BPA was 37 months. According to the decisions of the multidisciplinary team, of our 64 patients 47 (73.4%) were considered inoperable, 4 (6.3%) had technically operable disease but they refused surgical treatment and 13 (20.3%) had residual PH after PEA. Furthermore, we compared patients by class of their anticoagulant (warfarin vs. NOACs) and observed significant difference in median time from diagnosis (59 vs. 7.5 months; *p* = 0.020) and median baseline NTproBNP level (458.0 vs. 984.5 pg/mL; *p* = 0.044); also proportion of inoperable patients (63% vs. 89%) was highly distinct. In total, we treated 749 target lesions during 160 BPA procedures. With respect to target lesion classification, the following categories were determined: 82.2% ring-like stenosis and web lesions, 5.6% subtotal lesions, 9.9% total occlusion lesions and 2.3% tortuous lesions. More details are summarized in Table 1.

Up to 31 October 2019, 25 patients (56% of females, mean age of 63 ± 10.5 years) completed their BPA series. Overall, 74 BPA treatment sessions (305 segmental branches in total) were performed according to criteria established for the initial phase of the BPA developmental program for completion of BPA intervention. This subgroup for analysis of efficacy endpoints consisted of 18 (72%) inoperable patients, 2 (8%) technically operable patients who refused PEA and 5 (20%) patients with residual PH after PEA. Further, 14 (56%) patients had received specific vasodilator therapy at least 3 months before BPA (10 riociguat, 2 subcutaneous treprostinil and 2 endothelin receptor antagonists). The median (5th; 95th percentile) timepoint when data were collected to evaluate status after the completion of intervention was 4.0 (2.0; 14.0) months after the last BPA.

### 3.2. Effect of BPA on Clinical Outcome

Before the first BPA, NYHA classes II, III and IV were observed in 4.2%, 91.7% and 4.2% of patients, respectively. After completion of the BPA series, NYHA classes I, II and III were observed in 4.2%, 75% and 20.8% of patients, respectively (Figure 3). Moreover, NYHA class decreased by at least one degree in 75% of patients.

To further investigate patient characteristics associated with successful BPA outcome, we compared all baseline parameters before the first BPA in subgroup with NYHA class decreased by at least one degree (18 patients) vs. subgroup with no improvement in NYHA functional class (6 patients). Except mean (SD) pulmonary artery systolic pressure (73.7 (17.7) mmHg vs. 54.2 (11.8) mmHg; *p* = 0.015), the other parameters were not significantly different between both subgroups.

The following most important significant improvements were observed after completion of the BPA series in comparison to baseline values: 6MWD increased by +54.3 ± 48.3 m (*p* < 0.001), PAMP decreased by 18% (*p* < 0.001), PVR decreased by 32% (*p* < 0.001), stroke volume increased by 17% (*p* = 0.011), cardiac efficiency increased by 46% (*p* < 0.001), compliance increased by 38% (*p* < 0.001) and elastance decreased by 27% (*p* < 0.001). NTproBNP level decreased by 26% of baseline value (*p* = 0.011). Table 2 shows the results of all efficacy endpoints in detail.

### 3.3. Effect of BPA on Risk Profile

All three noninvasive risk variables (NYHA functional class, 6MWD and NTproBNP) were available in 19 patients at both time-points (before the first BPA and after the last BPA session). Before the first BPA 52.6%, 31.6%, 10.5% and 5.3% of them had no, one, two or three low-risk criteria, respectively. After the last BPA session 15.8%, 15.8%, 47.4% and 21.1% of them had no, one, two or three low-risk criteria, respectively (Figure 4). The risk score was unchanged in 21.1% patients and improved in 52.6% and 26.3% in 1 or 2 categories, respectively (*p* < 0.001).

### 3.4. Effect of BPA on Quality of Life

Quality of life evaluated by patients themselves before the first and after the last BPA improved significantly as measured by the change of EQ5D-5L score (0.20 ± 0.12; *p* < 0.001) and overall health status (37.0 ± 15.4%; *p* < 0.001).

### 3.5. Safety

Out of 160 BPA sessions, we observed one fatal complication (1.6% of all patients) when a non-cardiac pulmonary oedema emerged within 12 h after the first BPA session, and despite extracorporeal membrane oxygenation (ECMO) support the patient died 6 days later. This patient was the third who underwent BPA at our center when larger balloons were used during the intervention at the time. Moreover, he had systemic pressures in pulmonary artery with proximal lesions and he was on chronic hemodialysis.

In total, 19 BPA-related non-fatal complications occurred during 160 procedures (11.9% of all interventions). We detected predominantly hemoptysis (10.0% of all sessions), followed by single cases of vagal reaction (0.6% of all sessions), atrioventricular (AV) block III (0.6% of all sessions) and aspiration (0.6% of all sessions).

In the population with completed BPA, mean serum creatinine levels did not change after the BPA series (89.8 ± 21.1 μmol/L before the first BPA and 90.3 ± 22.9 μmol/L after the last BPA).

### 3.6. Survival

In total, there were five (7.8%) deaths among 64 patients with at least one BPA session. Only one periprocedural death (1.6%), as described in the section above, occurred within 30 days after BPA. No patient underwent lung transplantation.

Overall survivals (95% confidence interval) at 6, 12, 18 and 24 months were 96.6% (92.2–100.0%), 94.6% (88.8–100.0%), 87.0% (76.2–99.3%) and 87.0% (76.2–99.3%), respectively (Figure 5). No difference in survival was observed in subgroups by class of administered anticoagulant (warfarin vs. NOACs).

## 4. Discussion

Here, we demonstrated the favorable safety of the BPA procedure in 64 patients who underwent at least one BPA session—160 BPA sessions in total. Furthermore, we confirmed the efficacy of the procedure in a subgroup of 25 patients with completed BPA, with significant improvements in NYHA functional class, 6MWD, hemodynamics and right ventricular function. Likewise, patients reported significantly better quality of life after the BPA procedure was completed.

The BPA is a very important method of CTEPH treatment since even in specialized CTEPH centers up to 50% of patients diagnosed with CTEPH cannot undergo PEA (as being technically inoperable, medically inoperable or refusing surgery). Moreover, a substantial proportion of patients suffer from persistent PH after PEA. Specifically in the General University Hospital in Prague, 42.2% of CTEPH patients do not undergo PEA and 30% of patients after PEA have residual PH. In this study, the spectrum of patients indicated for BPA reflects these proportions, with 79.7% of patients being inoperable or refusing PEA and 20.3% with residual PH after PEA. Similar numbers (91.8% inoperable or refusing PEA, 8.2% with residual PH after PEA) were reported from one of the largest CTEPH centers in the world with PEA and BPA programs in Paris [14].

We would like to stress that BPA is not a competitive treatment option for the typical proximal obstructive lesions arising from non-resolving thromboemboli operable by PEA surgery. Nevertheless, technically operable patients might be candidates for BPA if the risk of the surgery is unacceptably high or if they refuse the surgery. In our cohort, four (6.3%) patients with BPA were considered as technically operable, but they had refused PEA. Main target lesions for BPA in these patients are not typical proximal obstructive operable lesions but predominantly more distal lesions that are also present in patients with proximal lesions, where for example rupture of the web after balloon inflation leads to improvement of blood flow.

The European Society of Cardiology guidelines from 2015 [1] states that inoperable patients and patients with residual PH after PEA should receive targeted medical therapy. In recent years, improvement in BPA technique has resulted in better clinical outcome and lower complication rates [10]. Therefore, patients, in whom only minor peripheral vascular remodeling is expected, might be indicated for BPA even without previous medical therapy. This approach is based on decision of interdisciplinary CTEPH team selecting treatment strategy for each patient individually. In our study, 44% of patients with completed BPA had not received any specific vasodilator therapy at least 3 months before the first BPA. For comparison, 38% of patients in French cohort [14] and 28% of patients in Japanese registry [10] had not been treated with targeted medical therapy before BPA.

### 4.1. Effect of BPA on Clinical Outcome

Balloon pulmonary angioplasty efficacy observed in our study was similar to results achieved in the currently largest set of European patients (184 patients) from France [14]. Proportions of patient with functional class NYHA III or IV were 96% at baseline and 20.8% after the last BPA in our cohort, while in the French study they were 64.7% at baseline and 21.3% after the last BPA. Mean 6MWD amelioration of 54 m in our subgroup is equivalent to the French increase of 45 m. With respect to hemodynamics, we achieved a decrease in PAMP of 18% and in PVR of 32%, while French group reported 26% and 43%, respectively. Nevertheless, each patient in the French subset underwent, on average, five to six BPA sessions while in our study only three BPA sessions. Results from other European centers where BPA programs are implemented are similar. For example, two sites in Germany performed 266 BPA sessions in 56 patients and achieved a 6MWD amelioration of 33 m, PAMP decrease of 18% and PVR decrease of 26% [13].

Results from a Japanese registry conducted in seven experienced BPA centers show significantly higher BPA efficacy, although all institutions in Europe and the USA have recently adopted the technique perfected in Japan [10]. This difference in efficacy may be due to the higher number of sessions per patient in Japanese centers, broader experience and possibly different patient characteristics. In Europe, over 50% of CTEPH patients undergo PEA, so only patients who are not eligible for PEA are predominantly indicated for BPA. In contrast, PEA is relatively unavailable in Japan and only 19.5% of patients are considered as technically operable. Therefore, some patients who undergo BPA in Japan are considered operable by PEA in Europe and the USA. About 80% of Japanese patients undergoing BPA are females, compared to about 50–60% in Europe. Different vascular phenotypes of European and Japanese patients with CTEPH could be also responsible for the different effects of BPA series in Japanese populations [20]. Moreover, the time from CTEPH diagnosis to the first BPA session could play an important role. In a large set of 500 interventions in 97 patients in Japan, the mean time from CTEPH diagnosis to the first BPA was 32.7 ± 36.8 months [6] while it was 48.8 ± 46.8 months in our study. Protracted disease duration brings a risk of deterioration in peripheral vascular remodeling that, in consequence, increases PVR not manageable by BPA.

Multiple RV function parameters (stroke volume, stroke volume index, cardiac efficiency, pulmonary artery elastance and compliance) were significantly improved in our cohort after the last BPA session. Furthermore, NTproBNP as a marker of RV stretch was significantly reduced. Right ventricular function is a critical outcome determinant in patients with PH. An association between RV function and survival has already been described in patients with PAH treated with specific therapy [21,22,23]. We speculate that this could be also applicable for CTEPH patients treated with BPA. Nevertheless, we are not able to confirm this association in our cohort since the sample size was rather small and no death occurred within the subgroup of patients with completed BPA.

Oral vitamin K antagonists are recommended to patients with CTEPH. Despite a lack of safety and efficacy data of the NOACs use in the CTEPH population, the number of patients with CTEPH, who use NOACs, is increasing. In our cohort mainly technically inoperable patients and patients with shorter period between the diagnosis of CTEPH and the first BPA received NOACs. Unfortunately, we were not able to analyze potential differences in clinical outcome after completed BPA in subgroups treated with warfarin vs. NOACs due to small sample size.

### 4.2. Effect of BPA on Risk Profile

Risk assessment at diagnosis and its monitoring during follow-up is currently widely used in patients with PAH. Methods of risk stratification have been already validated in several real world cohorts. Boucly et al. [18] proposed three noninvasive criteria (WHO/NYHA functional class, 6MWD and BNP/NTproBNP) for the accurate prognosis predictions in PAH patients. Delcroix et al. [24] demonstrated an association between risk assessment and survival in medically treated CTEPH patients from the real-world registry Comparative, Prospective Registry of Newly Initiated Therapies for Pulmonary Hypertension (COMPERA). Moreover, they also demonstrated that the number of present low-risk noninvasive criteria was associated with survival. No low-risk criteria were reported in 64.1% of patients at baseline and in 47.7% of patients at 3 months to 2 years follow-up.

In our cohort of CTEPH patients after the completed BPA series, we demonstrated a significant improvement of their risk profile assessed by three noninvasive variables (*p* < 0.001). No low-risk criteria were reported in 52.6% of patients at baseline and only in 15.8% of patients after the last BPA session. This is much better result in comparison to patients without specification of the CTEPH treatment method in COMPERA registry. To our knowledge, this is the first study that reports significant improvement of risk profile in CTEPH patients after completed BPA treatment.

Since no death occurred in our subgroup of patients with a completed BPA series, we are not able to analyze the association between risk profile and survival.

### 4.3. Effect of BPA on Quality of Life

The quality of life in patients with untreated chronic PH is poor, usually similar to the quality of life in patients with advanced stages of pulmonary, cardiac or renal disorders [25]. In our study, patient quality of life significantly improved after completed BPA in comparison to baseline status (*p* < 0.001 for both EQ5D-5L score and overall health status). Of note, the impact of BPA on quality of life is not evaluated in majority of studies published to date.

### 4.4. Safety

Balloon pulmonary angioplasty is an invasive procedure that is associated with complications like pulmonary oedema, pulmonary vascular injury, vessel injury, pulmonary hemorrhage or hemorrhagic pleural effusion [26]. According to data from the Japanese multicenter registry, the overall rate of BPA related complications was 36.3%, with pulmonary injury and hemoptysis the most common complications; severe complications occurred in 5.5% of all patients [10].

In our study, we observed one fatal complication (i.e., in 1.6% of all patients) at the very beginning of our BPA program, before refined Japanese technique was adopted. Out of 160 BPA sessions, the rate of non-fatal BPA-related complications was 11.9%, with hemoptysis the most frequent event (in 10% of all BPA sessions). Our results correspond well with experience from other European centers. For example, French authors reported BPA complications in 15.8% of all sessions during the initial period of the program and 7.7% during the more recent period, and the mortality rate was 2.2% within 30 days after the procedure [14]. Two sites from Germany reported complications in 9.4% of all BPA interventions and a mortality rate of 1.8% [13].

Despite the potential risk of renal function impairment caused by repetitive administration of contrast media during BPA sessions, no significant change in serum creatinine from baseline was detected in patients with a completed BPA series. Of note, Kriechbaum et al. [27] described even a slight improvement in renal function in individuals with chronic kidney disease, probably due to improvements of pulmonary and systemic hemodynamics after BPA.

### 4.5. Survival

Untreated CTEPH is characterized by poor survival [28]. When PEA is applicable, it improves the survival significantly. In the International Prospective Registry with 679 newly diagnosed CTEPH patients, survival at 1 and 2 years was 93% and 91% in patients after PEA and 88% and 79% in non-operated patients, respectively [29]. Residual PH was confirmed as one of the main factors that increase mortality. Therefore, BPA in specialized centers with an established surgical program is primarily intended for CTEPH patients with suboptimal prognosis–inoperable patients and patients with residual PH after PEA.

Although long term survival after at least one BPA session from our study (survival at 1 and 2 years was 95% and 87%) could not be directly compared with the survival of patients included in the above mentioned registry due to the immortal time bias (as the registry reported survival from the time of diagnosis), our results are comparable with the survival data after BPA from the Japanese registry (survival at 1 and 2 years was 93% and 91%) [10].

### 4.6. Limitations

Our study presented here has several limitations. First of all, the number of included patients was relatively small and the results could be skewed by the learning curve. This is an interim analysis performed within the scope of our BPA program, with only 25 patients completing their BPA series and having data available for a BPA efficacy analysis. Secondly, no control group was implemented, but this arises from the nature of a real-life design. Thirdly, this is a single center study. Last but not least, in this initial phase of our BPA program, we collected data retrospectively. However, we plan to follow our patients prospectively in the next stages of our registry. Despite these limitations, we were able to demonstrate the important and complementary role of BPA in an experienced CTEPH center with a successful PEA program.

## 5. Conclusions

Balloon pulmonary angioplasty is an effective therapeutic option with acceptable safety that should be integrated as a standard part of the therapeutic spectrum in established CTEPH centers. Balloon pulmonary angioplasty provides further improvements of functional class, exercise capacity, hemodynamics, right ventricular function, risk profile and quality of life to patients who are not candidates for PEA or to patients with residual PH after PEA.

## Figures and Tables

**Figure 1 jcm-09-03608-f001:**
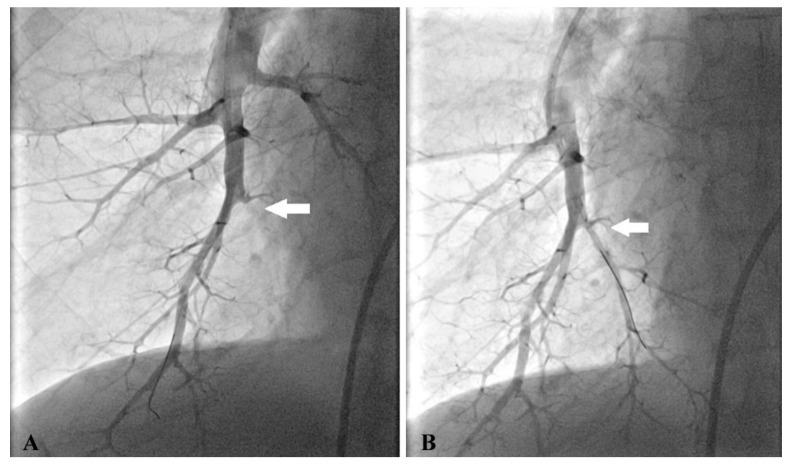
Selective pulmonary angiogram of the right lower lobe shows total occlusion of segmental branch A10 (**A**) and its opening after the first BPA procedure (**B**).

**Figure 2 jcm-09-03608-f002:**
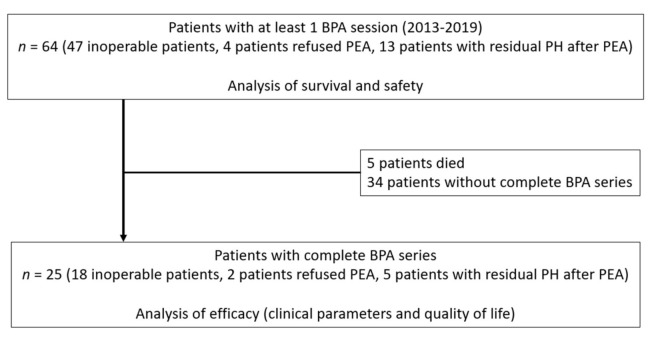
Study profile. Abbreviations: BPA—balloon pulmonary angioplasty; *n*—number; PEA—pulmonary endarterectomy; and PH—pulmonary hypertension.

**Figure 3 jcm-09-03608-f003:**
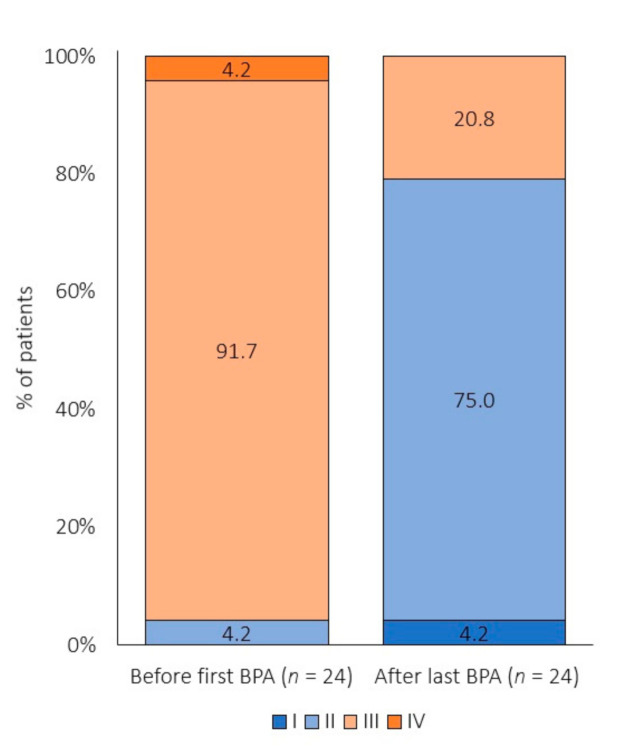
NYHA functional class before the first and after the last BPA session. Abbreviations: NYHA—New York Heart Association and BPA—balloon pulmonary angioplasty.

**Figure 4 jcm-09-03608-f004:**
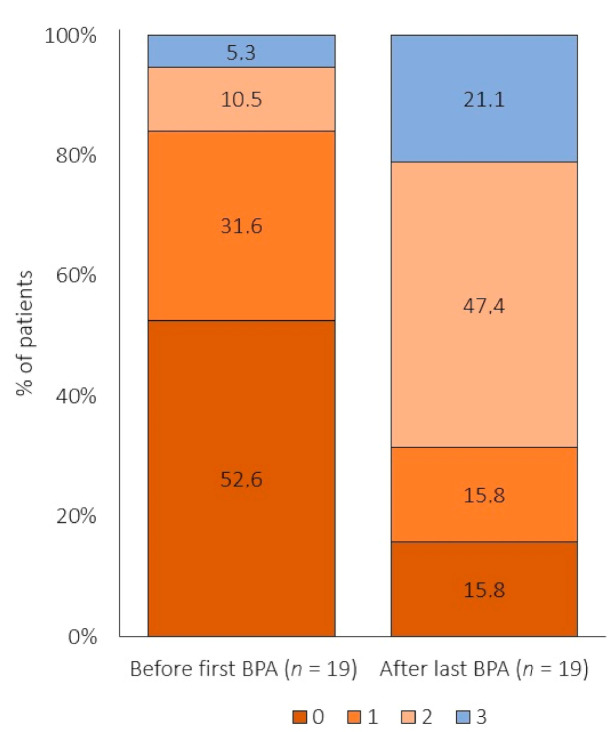
Risk profile assessed by the presence of three noninvasive variables (NYHA functional class, 6MWD and NTproBNP) before the first and after the last BPA session. Abbreviations: BPA—balloon pulmonary angioplasty; *n*—number; NYHA—New York Heart Association; 6MWD—6-min walking distance; and NTproBNP—N-terminal pro-brain natriuretic peptide.

**Figure 5 jcm-09-03608-f005:**
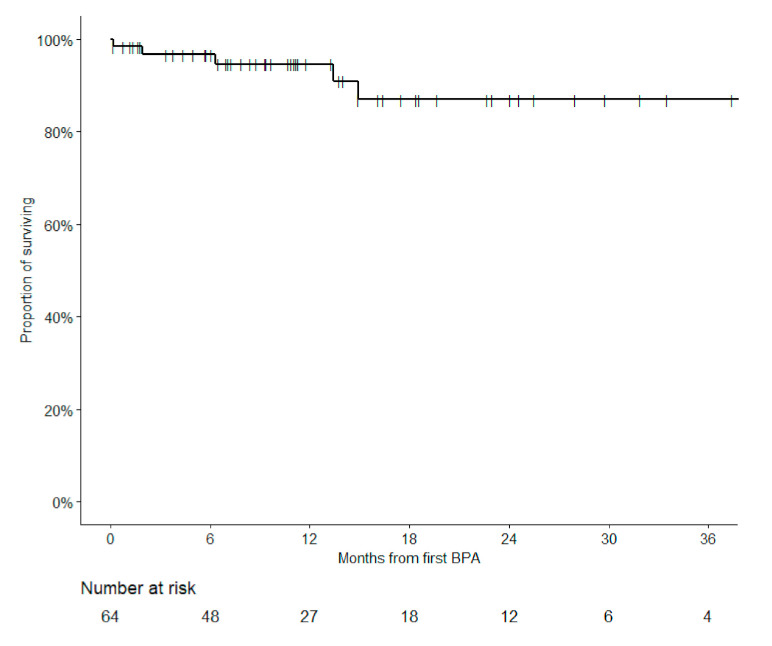
Overall survival of all 64 patients with at least one BPA procedure.

**Table 1 jcm-09-03608-t001:** Patient characteristics at the time of chronic thromboembolic pulmonary hypertension (CTEPH) diagnosis (*n* = 64).

Parameter		*n* (%)	Valid *n*	Mean (SD)	Median (5th; 95th Percentile)
Sex	Males	32 (50.0%)			
	Females	32 (50.0%)			
Age (years)			*n* = 63	61.4 (12.1)	62.0 (46.0; 78.0)
Height (cm)			*n* = 60	171.3 (10.4)	171.0 (154.5; 186.0)
Weight (kg)			*n* = 61	84.4 (16.9)	84.0 (60.0; 112.0)
BMI (kg/m²)			*n* = 59	29.0 (5.4)	28.4 (21.1; 39.6)
PE history		52 (81.3%)			
Hereditary thrombophilia		12 (18.8%)			
Diuretics		33 (51.6%)			
Anticoagulation *	Warfarin	41 (68.3%)			
	NOAC	19 (31.7%)			
FEV1 (% of predicted)			*n* = 54	89.3 (19.1)	89.5 (57.0; 119.0)
FVC (% of predicted)			*n* = 54	97.6 (19.3)	98.6 (57.0; 129.0)
6MWD (m)			*n* = 52	385.5 (110.1)	387.5 (207.0; 597.0)
Borg dyspnoea index			*n* = 51	5.8 (1.7)	6.0 (3.0; 8.0)
NYHA *	I	0 (0.0%)			
	II	16 (25.8%)			
	III	44 (71.0%)			
	IV	2 (3.2%)			
RAMP (mmHg)			*n* = 63	9.4 (5.1)	9.0 (2.0; 19.0)
PASP (mmHg)			*n* = 64	75.8 (18.8)	77.5 (45.0; 104.0)
PADP (mmHg)			*n* = 64	27.0 (9.2)	26.0 (13.0; 40.0)
PAMP(mmHg)			*n* = 64	44.0 (12.7)	44.0 (26.0; 62.0)
PCW (mmHg)			*n* = 64	9.9 (3.6)	10.0 (4.0; 15.0)
CO (L/min)			*n* = 63	4.5 (1.1)	4.5 (2.7; 5.8)
CI (L/min/m^2^)			*n* = 64	2.3 (0.5)	2.2 (1.6; 3.1)
PVR (WU)			*n* = 63	8.3 (3.9)	7.7 (3.3; 16.4)
HR (beats/min)			*n* = 61	71.3 (14.3)	70.0 (54.0; 97.0)
SvO2 (%)			*n* = 63	65.0 (8.5)	65.7 (51.5; 75.0)
TPR (WU)			*n* = 63	10.6 (4.5)	9.9 (5.0; 18.6)
SV (L/beat)			*n* = 61	0.066 (0.023)	0.065 (0.030; 0.098)
SVI (L/beat/m^2^)			*n* = 61	0.034 (0.010)	0.033 (0.018; 0.048)
Cardiac efficiency (L/beat/mm Hg)			*n* = 61	0.0017 (0.0009)	0.0015 (0.0006; 0.0033)
Compliance (L/beat/mm Hg)			*n* = 61	0.0014 (0.0007)	0.0013 (0.0006; 0.0026)
PA elastance (mmHg/L/beat)			*n* = 61	1338.3 (679.3)	1134.0 (567.5; 2490.0)
Creatinine (μmol/L)			*n* = 64	100.0 (80.4)	89.0 (65.0; 124.0)
BNP (pg/mL)			*n* = 21	341.2 (571.3)	159.0 (17.0; 619.0)
NTproBNP (pg/mL)			*n* = 31	1021.3 (1102.7)	590.0 (50.0; 3,032.0)

* Information on anticoagulation is missing in 4 patients. Information on NYHA class is missing in 2 patients. Continuous variables were described by valid number (*n*), mean (SD) and median (5th and 95th percentile). Absolute and relative frequencies calculated from valid number were used for descriptions of categorical variables. Abbreviations: *n*—number; SD—standard deviation; BMI—body mass index; PE—pulmonary embolism; NOAC—Non-vitamin K antagonist oral anticoagulants, FEV1—forced expiratory volume in 1 s; FVC—forced vital capacity; 6MWD—6-min walking distance; NYHA—New York Heart Association; RAMP right atrial mean pressure; PASP—pulmonary artery systolic pressure; PADP—pulmonary artery diastolic pressure; PAMP—pulmonary artery mean pressure; PCWP—pulmonary capillary wedge pressure; CO—cardiac output; CI—cardiac index; mmHg—millimeter of mercury; PVR—pulmonary vascular resistance; TPR—total pulmonary vascular resistance; HR—heart rate; WU—Wood units; SvO2—mixed venous oxygen saturation; SV—stroke volume; SVI—stroke volume index; PA—pulmonary artery; BNP—brain natriuretic peptide; and NTproBNP— N-terminal pro-brain natriuretic peptide pulmonary hypertension.

**Table 2 jcm-09-03608-t002:** Comparisons of clinical outcomes before the first and after the last BPA session in patients with completed BPA treatment (*n* = 25).

Parameter	Valid *n*	Before the First BPA		Absolute Diference ^1^		*p*-Value	Relative Diference ^2^	
		Mean (SD)	Median (5th; 95th Percentile)	Mean (SD)	Median (5th; 95th Percentile)		Mean (SD)	Median (5th; 95th Percentile)
NYHA	25	3.0 (0.3)	3.0 (3.0; 3.0)	−0.8 (0.6)	−1.0 (−2.0; 0.0)	<0.001	0.73 (0.18)	0.67 (0.50; 1.00)
6MWD (m)	22	369.0 (116.3)	381.5 (151.5; 527.0)	54.3 (48.3)	45.5 (−17.0; 146.5)	<0.001	1.19 (0.20)	1.16 (0.96; 1.63)
Borg dyspnoea score	22	6.3 (1.9)	6.0 (2.5; 8.5)	−1.4 (1.7)	−1.5 (−4.5; 1.0)	0.003	0.83 (0.27)	0.81 (0.35; 1.35)
NTproBNP (pg/mL)	24	1107.9 (1458.8)	482.0 (90.0; 3 778.0)	−535.9 (949.4)	−54.0 (−2 384.0; 60.0)	0.011	0.74 (0.38)	0.70 (0.24; 1.34)
Creatinine (mmol/L)	25	89.8 (21.1)	89.0 (62.0; 116.0)	0.5 (22.4)	2.0 (−30.0; 39.0)	0.554	1.03 (0.21)	1.03 (0.71; 1.39)
RA (mmHg)	24	7.3 (4.3)	8.0 (1.0; 13.0)	−1.7 (5.2)	−1.0 (−10.0; 5.0)	0.158	1.21 (1.36)	0.75 (0.20; 5.00)
PASP (mmHg)	25	69.8 (17.8)	68.5 (43.0; 100.0)	−11.7 (12.3)	−14.0 (−25.0; 11.0)	<0.001	0.84 (0.17)	0.78 (0.65; 1.15)
PADP (mmHg)	25	24.4 (8.9)	24.5 (12.0; 40.0)	−5.1 (7.0)	−6.0 (−17.0; 4.0)	0.004	0.83 (0.24)	0.80 (0.54; 1.17)
PAMP (mmHg)	25	39.7 (11.1)	37.0 (23.0; 58.0)	−7.4 (6.9)	−7.0 (−19.0; 5.0)	<0.001	0.82 (0.16)	0.78 (0.57; 1.14)
PCW (mmHg)	25	9.3 (4.0)	10.0 (3.0; 14.0)	0.5 (4.6)	0.0 (−6.0; 7.0)	0.542	1.39 (1.00)	1.00 (0.50; 3.67)
CO (L/min)	25	4.9 (1.2)	4.9 (3.6; 6.9)	0.4 (1.3)	0.0 (−1.3; 2.1)	0.187	1.11 (0.26)	1.01 (0.81; 1.48)
CI (L/min/m^2^)	25	2.5 (0.5)	2.5 (1.8; 3.7)	0.2 (0.6)	0.0 (−0.6; 1.1)	0.137	1.12 (0.26)	1.00 (0.81; 1.49)
PVR (WU)	25	6.6 (3.0)	5.9 (3.5; 11.1)	−2.2 (1.6)	−1.7 (−5.1; −0.3)	<0.001	0.68 (0.19)	0.71 (0.39; 0.95)
HR (bpm)	25	72.9 (11.3)	71.0 (54.0; 90.0)	−4.0 (10.2)	−3.0 (−23.0; 9.0)	0.107	0.96 (0.12)	0.95 (0.74; 1.14)
SvO2 (%)	23	63.8 (8.8)	65.0 (46.5; 75.0)	3.6 (6.7)	2.0 (−5.0; 17.0)	0.041	1.07 (0.13)	1.03 (0.93; 1.36)
SV (L/beat)	25	0.069 (0.020)	0.069 (0.041; 0.108)	0.009 (0.015)	0.006 (−0.013; 0.032)	0.011	1.17 (0.26)	1.09 (0.81; 1.57)
SVI (L/beat/m^2^)	25	0.035 (0.008)	0.035 (0.024; 0.049)	0.005 (0.008)	0.004 (−0.008; 0.018)	0.005	1.17 (0.25)	1.10 (0.79; 1.61)
Cardiac efficiency (L/beat/mmHg)	25	0.0019 (0.0008)	0.0018 (0.0008; 0.0033)	0.0007 (0.0005)	0.0006 (0.0002; 0.0013)	<0.001	1.46 (0.37)	1.40 (1.06; 2.09)
Compliance (L/beat/mmHg)	25	0.0016 (0.0006)	0.0015 (0.0008; 0.0025)	0.0006 (0.0004)	0.0005 (−0.0001; 0.0012)	<0.001	1.38 (0.31)	1.39 (0.97; 1.84)
PA elastance (mmHg/L/beat)	25	1097.5 (462.5)	1006.5 (580.1; 1 785.5)	−323.2 (265.9)	−225.8 (−786.5; −23.1)	<0.001	0.73 (0.15)	0.73 (0.48; 0.97)

Statistical significance of difference between both timepoints was analyzed by the Wilcoxon paired test. ^1^ Absolute difference was calculated as the difference between values from the last BPA and before the first BPA. ^2^ Relative difference was calculated as the quotient of the value from the last BPA divided by the value from the first BPA. Abbreviations: BPA—balloon pulmonary angioplasty; *n*—number; SD—standard deviation; NYHA—New York Heart Association; 6MWD—6-min walking distance; NTproBNP—N-terminal pro-brain natriuretic peptide; RAMP right atrial mean pressure; PASP—pulmonary artery systolic pressure; PADP—pulmonary artery diastolic pressure; PAMP—pulmonary artery mean pressure; PCWP—pulmonary capillary wedge pressure; CO—cardiac output; CI—cardiac index; mmHg—millimeter of mercury; PVR—pulmonary vascular resistance; HR—heart rate; WU—Wood units; SvO2—mixed venous oxygen saturation; SV—stroke volume; SVI—stroke volume index; and PA—pulmonary artery.

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
