# Peer review of "Balloon Pulmonary Angioplasty in Patients with Chronic Thromboembolic Pulmonary Hypertension: Impact on Clinical and Hemodynamic Parameters, Quality of Life and Risk Profile"

_jcm, 2020, doi:10.3390/jcm9113608_

Round 1
Reviewer 1 Report
the authors have adequately responded to my comments.
Author Response
We are grateful for the comments and suggestions.
Reviewer 2 Report
The aim of the study was to evaluate the safety and efficacy of balloon pulmonary angioplasty (BPA) in patients with chronic thromboembolic pulmonary hypertension (CTEPH). The authors concluded that BPA is an effective therapeutic method with acceptable safety which should be performed in experienced CTEPH centers. Ir provides improvement in functional class, exercise capacity, hemodynamics, right ventricular function, risk profile and quality of life in patients not elligible for PEA.
The paper is well-designed and well-written. However, I have the following concerns:
- Could you please transfer aim of the study to the introduction section?
- Please clarify whether patient informed consent form was obtained? If so please add it to the Method section.
- Please delete tracking changes
- Could you please add subanalysis including baseline characteristics and outcomes in relation to VKA/ NOAC?
- Figure 2 has low quality and it should be improved.
- Please do not begin a sentence with abbrevaition.
Round 2
Reviewer 2 Report
Thank you. All my concerns have been properly addressed. I have no further comments.
This manuscript is a resubmission of an earlier submission. The following is a list of the peer review reports and author responses from that submission.
Round 1
Reviewer 1 Report
The reviewer thanks the authors for their well written and quite interesting manuscript entitled “Balloon pulmonary angioplasty in patients with chronic thromboembolic pulmonary hypertension: impact on clinical and hemodynamic parameters, quality of life and risk profile”.
Since BPA gets widespread attention, national experiences are important.
There are only some minor remarks:
Experimental Section, 2.2:
Could you please describe your indication for BPA (target lesions) in more detail?
Furthermore, performing BPA in operable lesions can be quite challenging. Since this topic is also part of the discussion – the reviewer would like to ask for a critical statement: is an operable target lesion (usually a bigger amount of scar tissue) really a reasonable target for balloon angioplasty?
The reviewer is aware of the changes of the guidelines in the last couple of years. But meanwhile, targeted medical treatment is recommended in all inoperable CTEPH patients (and patients with persistent PH after PEA) – in the presented cohort, only 56 % of all patients underwent medical treatment (according to their PVR). This needs to be discussed and you should explain, which RHC has been the baseline (RHC at the time of diagnosis or RHC before the first BPA (like indicated in table 2).
Reviewer 2 Report
Jansa et al describe the outcome (clinical, hemodynamic parameters, quality of life etc) of balloon pulmonary angioplasty in 64 patients.
This is a nice report and the authors should be congratulated for their efforts. However, several issues need to be addressed.
- What is the novelty of this study compared with other previous reports. We already know that the procedure, if done right, significantly improves hemodyamic profiles and especially symptoms in CTEPH patients.
- The number of patients enrolled (64 total and ony 25 who completed BPA series) make it difficult to make any confirmatory conclusions.
- Would it be possible to differentiate from their data, which patients have the greatest improvement in symptoms? Which surrogate hemodynamic markers are best associated with clinical improvement. (Just a descriptive narration of the outcome and phenomenon, unless it is one of the first reports, have minimal scientific merit)